# Dark Theology as an Approach to Reassembling the Church

**Andrey Shishkov**

School of Theology and Religious Studies, University of Tartu, 50090 Tartu, Estonia; andrey.shishkov@ut.ee

**Abstract:** Dark theology as a theoretical approach emerged during debates on human rights and inclusion in Orthodox theology. It is realized at the junction of such disciplines as ecclesiology, political theology, philosophy, and social theory. It is based on the tools of object-oriented ontology (OOO), one of the branches of the philosophy of speculative realism. The author proposes a theoretical framework by which we can talk about God and supernatural entities as real objects included in public discourses through the collective imagination. The article discovers the basic theoretical (ontological, epistemological, and aesthetic) principles of dark theology as they apply to ecclesiology and political theology. Additionally, it discusses the existence of church dark actors who do not come within the field of vision of the theological mind (ecclesiology) illuminating ecclesial space. The author concludes by proposing a concept of reassembling the Church based on Bruno Latour's notion of the 'collective'.

**Keywords:** dark theology; object-oriented ontology; Orthodox theology; ecclesiology; Bruno Latour; Graham Harman; inclusion; political theology

## 1. Introduction

The French philosopher and sociologist Bruno Latour once wrote: "Religion is the opium but not of the people, it is the opium that puts social scientists readily to sleep at the very moment when those they are in charge of studying are being made to act by others" (Latour 2001, p. 230). Sociologists, he said, dream that they provide explanations more apt than those given by the actors themselves for why they act or why they are acted upon (Ibid). When any religious actor says that his actions were motivated by God or some other spiritual entity, or that s/he influences people by appealing to supernatural things (for example, by magic, ritual, etc.), sociologists deny the reality and objectivity of these things, attributing them to the work of imagination, the subconscious, or calling them projections of social relations, etc. The same problem often arises in Christian political theology, where God and supernatural entities are thought of only as symbols or analogies to justify human political processes and not as real actors (e.g., Carl Schmitt, Giorgio Agamben et al.).

The other extreme in political theology is to believe that God or grace is directly behind what happens in the world and give them more reality than the world itself. Only God possesses true existence, while creation is only relative and so on. At its extreme, this takes the form of 'naive spiritualism', where spiritual entities are understood as participants of life processes (including social ones) in the same sense as human beings. While sociologists fall asleep, theologians of this type fall prey to euphoria under the influence of the same "opium". The roots of these two extremes can be found, on the one hand, in Marxist or Durkheimian social reductionistic methodology, and on the other hand, in Platonic idealistic metaphysics.

However, even in approaches that stand between these two extremes, which, on the one hand, insist on interaction with God and, on the other, do not make him a direct participant in political processes, this interaction proves to be unclear. As Orthodox theologian Aristotle Papanikolaou writes, "the burden to figure out how to tap the latent divine presence within creation ultimately depends on humans" (Papanikolau 2012, p. 2). However, he stops short of this task, defining divine–human communion (a key concept of

his political theology) as "to love God with all of one's heart, soul, strength, and mind, that is, singularly, love the neighbor as self" (Ibid., p. 3). This is a rhetorical formula that has little to do with human communication with God. Thus, the term can be understood both as a metaphor for inter-human communication (first extreme) and as a description of direct interaction, in which one can even receive orders from God as a kind of spiritual guru or ruler (second extreme). However, of course, the author means neither the first nor the second. For him, divine–human communication is real but not reducible to direct contact.

All these positions force theologians to ask anew the question of how the presence of God and supernatural (spiritual) entities should be described in public discourse. On the one hand, it is necessary to avoid ignoring them completely, and on the other hand, it is necessary to evade the position that we have a direct access to supernatural objects in our experience.

My approach, which I call dark theology, is an attempt to evade these two extremes and to propose a theoretical framework by which we can talk about God and supernatural entities as real objects included in public discourses. I first used the term in December 2019 as applied to ecclesiology at a conference on philosophical theology at Russian State University for the Humanities. The first academic publication on dark ecclesiology came out in 2021 (Shishkov 2021). I have subsequently expanded this concept to dark theology (particularly in this publication and in the article for the Russian *Art Journal* to be published shortly).

Dark theology as a theoretical approach emerged during debates on human rights and inclusion in Orthodox theology. It is realized at the junction of such disciplines as ecclesiology, political theology, philosophy, and social theory. It is based on the tools of object-oriented ontology (OOO), one of the branches of the philosophy of speculative realism.

## 2. The Epistemological Problem

To avoid 'naive spiritualism', we should ask how we can know the supernatural, spiritual, or divine. The traditional cataphatic and apophatic methods of Christian theology give us some basic principles for knowing God. We can talk about his properties and mediated manifestations through the phenomena around us, nature, historical events, etc., but we cannot know him directly. These principles are summarized in Orthodox theology by the Palamite formula, which says: God is not cognizable in his essence but is cognizable in his energies (actions). However, this formula passed into oblivion for centuries. Furthermore, the Kantian turn in philosophy, which influenced theology as well, problematizes this very attitude: post-Kantian thinking defines all objects through thinking about them. In the 21st century, a philosophical direction emerged that questioned the entire modern and postmodern philosophical mainstream. It is called speculative realism.

In 2007, a workshop at Goldsmiths (University of London) brought together four philosophers who were strongly dissatisfied with the direction continental philosophy had taken since Kant. Their names were Quentin Meillassoux, Ray Brassier, Ian Hamilton Grant, and Graham Harman (founder of object-oriented ontology, essential for my approach). They laid down the basic principles of speculative realism, which then disintegrated into four different directions (one for each philosopher). Speculative realists sever the correlation, imaginary from their point of view, between the world and thinking about it. The term 'correlationism' proposed by Meillassoux denotes a standpoint from which "there is no possibility of considering thought or world in isolation from each other, since they are always treated as a pair existing only in mutual *correlation*" (Harman 2018b, p. 3). According to these philosophers, post-Kantian modern philosophy has made the existence of the world dependent on human thought. Speculative realism, first, assumes that human thought is finite, meaning that there are things we cannot think. Second, it argues that things are inherently withdrawn from cognition: consciousness is left to deal with the representations it has created. Third, these representations are contingent, unstable, and fragile: we do not and cannot have solid knowledge. Harman explains that withdrawal does not refer to some needlessly mystical disappearance of things from the immanent space but is simply

another way of saying that a form can exist in only one place; it cannot be moved—into a mind or anywhere else—without being translated into something different from what it was (Harman 2018b, pp. 104–9).

The peculiarity of the object-oriented approach, on which I rely, is that it insists on the priority of objects under all these conditions. They exist regardless of whether we can think them or not. For example, black holes in the universe existed before scientists discovered them, first theoretically and then practically. This is what a break with correlationism looks like. Similarly, spiritual entities as objects exist not because one thinks of them but autonomously. That said, we will never have direct access to them or complete knowledge about them.

Harman's theory of objects is based on a critique and then synthesis of two approaches: Edmund Husserl's phenomenology and Martin Heidegger's instrument-analysis. Harman's object analysis involves a distinction between real and sensual objects and their real and sensual qualities. The sensual object exists exclusively in our experience (it could be called an intentional object, but Harman deliberately avoids this definition). It is given to us directly, while the real one is withdrawn. It is the same with qualities: the sensual (accidental) qualities exist directly in our experience, the real (eidetic) ones are grasped indirectly by means of reason. The tension between these four 'poles' creates Harman's quadruple object (Harman 2011).

The real object emanates its sensual qualities into the realm of the present; this is the only means by which it enters our experience. We, as real objects, do not come into direct contact with other real objects. However, the trouble is, we do not come into direct contact with sense qualities either. For example, black does not exist as an isolated quality, but only as the black of ink. The bridge between me as a real object and the sensual qualities of another real object becomes another sensual object (you need ink to experience black). Harman rejects the possibility of direct access to objects, which natural science on the one hand and religious mysticism on the other both claim.

For example, from this perspective, if God is a real object, then the divine energies that exist in the mystic's experience as an experience of light are sensual qualities. Mystics contemplate God through communion with non-created divine energies, but what they see is a sensual object created in their minds.

It should be noted that Harman does not assert the reality of all objects. His idea is that real and unreal objects should be treated equally as objects. Moreover, fictional objects can have as much effect on reality as real objects. That said, we cannot exactly say that those objects we consider fictional do not exist in reality. For example, scientists used to believe that black swans did not exist, but this had no effect on the existence of these swans, which were eventually discovered. Even more obviously, this statement applies to hypothetical objects in the natural sciences, such as superstrings, gravitons, the multiverse, etc. This can also include spiritual entities that exist in the experience of individuals but cannot be detected through scientific observation.

## 3. Dark Ontology

OOO describes the finitude of our cognition by the metaphor of the 'darkness'. It was proposed by another philosopher belonging to this direction, Timothy Morton. He writes that "we live in a universe of finitude and fragility, a world in which objects are filled and surrounded by mysterious hermeneutic clouds of ignorance" (Morton 2016, p. 6). Dark ecology, according to Morton, is ecological consciousness, characterized by him as 'dark-depressing', 'dark-uncanny', and 'strangely dark-sweet' (Ibid., p. 5). This type of consciousness arises from the alarming insufficiency of our ways to describe the world around us, especially the scientific ones, which claim to be the main explanatory strategies of the modern world. From the perspective of dark ecology, the objects it studies become weird, and ecological consciousness becomes dark when confronted with them. It cannot penetrate 'deep into' the objects because they are withdrawn from cognition. The three characteristics of ecological consciousness described above are ways of encountering weird objects: the

perceived imperfection of describing the world around us is oppressive, frightening, and, at the same time, strangely appealing. The weirdness of things causes anxiety and attempts to get rid of it lead to violence through the imposition of explanatory strategies, which turn out to be just as imperfect (Ibid., p. 78). Morton describes this process as a loop, a vicious circle, or ouroboros. Morton's methodology has been applied to fields ranging from ecology to urban studies, in a sense becoming a universal epistemology of the strange and the dark. I call my project a dark theology in the same sense of epistemological darkness.

Religious consciousness also becomes 'dark' when confronted with the supernatural: first falling into fear (a frequent story in the Bible), and then finding it incomprehensibly appealing. The language with which this consciousness describes its experience is a dark apophatic language. As American theologian Catherine Keller writes, dark apophatic language creates a "state of alert in-betweenness and critical non-knowingness" (Keller 2003, p. 204). 'Critical non-knowingness' is not a rejection of knowledge in general but an awareness that this knowledge is limited and incomplete. She believes that the Christian mainstream is characterized by a fear of the metaphor of the dark. Keller argues that the demonization of the dark underlies modern Christian civilization. She describes the logic of this demonization as follows: "God called light 'good', so darkness must be 'bad'" (Ibid., p. 200). This demonization of the dark is an ouroboros in Morton's sense, a strategy for ridding ourselves of the disturbing incompleteness of our understanding of the supernatural.

The Christian theological language of "light, whiteness, and purity", from Keller's perspective, generates a language of binary oppositions in which one (light) subordinates the other (dark). Moreover, it reinforces existing binary oppositions. For example, traditional Christianity subordinates the feminine to the masculine (the most famous binary opposition) through its discursive ascription of the feminine to natural impurity, darkness, sinfulness. This emphasis on feminine purity/impurity is quite characteristic of the exclusive status of the Virgin Mary, who is called the pure one. There is no mention of the purity of Jesus or the apostles, for example. The demonization of the dark is characteristic not only of gender but also of race. In various catechisms of the past, one can read that the first dark-skinned man was Ham or even Cain. An entire race is stigmatized by attributing dark skin to biblical symbols of sinfulness. For example, in the ancient paterikons, one finds descriptions of demons associated with dark-skinned people. Keller even says that modern racism is a product of a certain type of Christianity. The dark becomes, by definition, demonic, sinful, evil, heretical.

Keller proposes to rehabilitate the theological language of the darkness because it provides that very "third place" (a term of postcolonial studies) that is beyond binary oppositions. She rightly points out that the darkness associated with God is present in the biblical narrative, theological reflection, and Christian mystical experience (e.g., Gregory of Nyssa, the Areopagite, Nicholas of Cusa et al.).

Divine darkness becomes a key concept for my approach. However, at the same time, dark theology problematizes the category of mystical experience, setting the conditions for the real encountering of God. The classical apophatic approach in theology does not distinguish between the degrees of darkness involved in knowing the divine. Contemporary philosopher of horror Eugene Thacker analyzed the concept of divine darkness in the texts of the Areopagite, Meister Eckhart, Angela of Foligno, the anonymous treatise "Cloud of Contemplation", John of the Cross, and Georges Bataille and distinguished three types of darkness that theologians and mystics deal with (Thacker 2015).

The first is *dialectical darkness*. Here, the concept of darkness is inseparable from its opposite—light; it is structured around the dyad dark/light and is found in such dyads as knowledge/ignorance, presence/absence, etc. In this kind of darkness, the movement goes from denying to affirming the experience of the divine, from the absence of any experience at all to full experience. This experience is at the same time affirmed through implicit negation: vision, which is at the same time blindness; ecstasy as an experience of the self-outside.

The second, *superlative darkness*, transcends all attempts to know the divine directly and is alien to empiricism, on the one hand, because it lies beyond the experience of light and dark, and to idealism, on the other hand, because it lies beyond the concept of light and dark. This darkness is described by contradictory concepts such 'luminous darkness' (Gregory of Nyssa), 'a ray of divine darkness' (the Areopagite). This type of darkness brings to the limit not only language, but also thinking, and posits a horizon for any possibility of thinking the impossible. The limit of human knowledge becomes a kind of limit to know the human. However, it is still possible to think (contemplate) that something exists, even though this something may be unknown to us. Both approaches do not give direct access to the divine and deal with sensual objects of the consciousness.

Finally, the *divine darkness* goes beyond the human limit. Here, there is still the assumption of something external, which turns out to be the limit for us as human beings. Such darkness does not provide the comforting knowledge of the unknown; rather, it is the knowledge that there is nothing to know. An encounter with something exceptionally inhuman.

Neither seeing nor hearing is possible in this darkness since both ways of knowing are mediated. However, there remains one other mode of interaction between objects: touch. The Orthodox theologian John Panteleimon Manoussakis writes that touch is "only sense that lacks a proper medium, and even more it operates by traversing anything mediating between the tangible and the tactile" (Manoussakis 2007, p. 4). However, without seeing and hearing, touching does not guarantee that you encounter the one you hope to discover. This experience requires a great deal of faith and courage. Thus, real mystical experience is fragile and unstable, giving only a presumed knowledge of God that occurs through direct touch, bypassing contemplation, and hearing. Any other forms of mystical experience deal with the experience of sensual objects in the mystic's mind.

## 4. Aesthetics and Politics

Dark theology, however, differs somewhat from the classical apophatic approach and Kantian noumenal. The appeal to divine darkness shows the fundamental impossibility of knowing God through (enlightened) reason. However, this does not mean that cognition is fundamentally impossible.

Morton believes that art becomes a strategy for describing the dark in an epistemological sense because "beauty gives you a fantastic, 'impossible' access to the inaccessible, to the withdrawn, open qualities of things, their mysterious reality" (Morton 2018, p. 41). Harman also points out the importance of aesthetics. He believes that this access is possible through the tools of aesthetics—for example, through metaphor. The metaphor works because of the deep divide between an object and its qualities, fusing the object with the qualities of another object to allow us to penetrate deeper into its understanding (Harman 2018a, p. 75). For example, in the phrase "bloody dawn", dawn is endowed with the qualities of blood, but this surprisingly gives us some additional knowledge of the object. Aesthetics works on the tension between real objects and sensual qualities. So, Gregory of Nyssa's luminous darkness becomes a way of penetrating the essence of the mystery of God, which cannot be described through a literal understanding of notions.

At the same time, since no other real object is available to us, the metaphor fuses sensual qualities with the only real object available to us: ourselves. In other words, to work, the metaphor must be lived by us from within. Each of us will have a different knowledge and degree of penetration into the object called "bloody dawn" or "luminous darkness".

Turning to theo-aesthetics, we can agree with David Bentley Hart that "beauty is the beginning and end of all true knowledge" (Hart 2004, p. 132) and that "Christian beauty is also a hidden beauty, prior to all "essentialist" representations, a messianic secret, a kenosis" (Ibid., p. 148). Despite this, he believes in the obviousness of the given, which he calls "the covenant of light", "a way of seeing that refuses to see more—or less—than what is given" (Ibid., pp. 145–46). The problematic aspect of this position is that he considers the aesthetic experience of God to be universal for any person. That is, different people "see"

the same God and have direct access to him as a real object. "The covenant of light", in fact, destroys the thesis of the hiddenness and mystery of God, hiding in the darkness of the hermeneutic clouds of ignorance.

However, all the real objects, including God, are not given to us directly, they are in the darkness. Only the sensual objects of our experience are given to us. In the case of aesthetic cognition, the evidence of givenness is achieved through the givenness to us of a single real object—ourselves.

When we talk about God or Christ, we are talking about their images in our minds—our visions of them, which turn out to be different for everyone. As a character in Jim Jarmusch's *Dead Man* movie called Nobody says: "The vision of Christ that thou dost see is my vision's greatest enemy". Everyone imagines their own Christ in the Gospel. Hart's covenant of light thus becomes a substitute for the real Christ, who continues to be in the shadows. The real Christ is the dark Christ. Additionally, the light Christ is us, fused with the qualities of Christ. It does not mean, of course, that Christ as a real object is removed from the world, only withdrawn from the direct access of our consciousness. An analogy here is the notion of an "imagined community" proposed by Benedict Anderson (Anderson 1991). The nation as an imagined community is real, yet it is withdrawn from cognition as a real object. We do not see the nation, only imagine our identification with it. However, this collective work of the imagination creates a political community.

Just as with the nation, God as a political actor is present in public space through his representation in people's imaginations. Thus, for example, the divine–human communion, Papanikolaou writes, is a metaphor in Harman's sense. Additionally, the ascetic struggle that becomes the primary driver of Papanikolaou's social ethics is the constant correlation of oneself with the qualities of God. It cannot be ruled out that God may or may not act in the world, but we have no direct access to the knowledge of this.

## 5. Dark Ecclesiology

However, it is not only God that could be dark. This methodology can also be applied to the Church because it is inhabited by a multitude of dark actors who do not come within the field of vision of the theological mind illuminating ecclesial space.

The existing ways of describing the Church in Orthodox ecclesiology are clericocentric. Through them, the rest of the Church views clerics as a chosen part of the Church people, whose priesthood gives them advantages not only of a practical nature but also, in some interpretations, of an ontological nature (ordination changes the nature of a person). Ecclesiologies describe the Church in such a way that clerical structures inevitably become their focal points and replace the Church's image. When we talk about the Church in everyday life, we immediately imagine a clergyman, worship, or church building. These ecclesiologies contain the message that if a person belongs to the right jurisdiction, participates rightly in the right style of worship and sacraments, follows the right practices, and correlates his faith with Orthodoxy—the content of which is also controlled by the clerics—then he will be saved. Such ecclesiological concepts as schism, heresy, Eucharistic communion, etc., become instruments of power control. Even the place of women in the church is discussed mainly in a clerical manner as the topic of female priesthood.

Through ecclesiology, church authorities presuppose a certain norm in the "church organism", implying that some ecclesial actors are oppressed or even excluded from the Church by them. At the same time, despite this "normative violence", the excluded do not disappear from the Church. They continue to exist in the shadow of the Church as something strange, undesirable, or simply inconvenient. Women, homosexuals, transgender and other queer people, unbaptized babies, human embryos, animals and other non-human beings, sacred objects and things, and cultural heritage are relegated to the ecclesiological shadow. It is hard to find them in ecclesiological descriptions, but their impact on church life is significant. It could be compared to dark matter in the universe.

The dark frightens with its incomprehensibility and uncertainty, and we are encouraged to get rid of it, to expel it from the Church, and not let it in to get to know it better. For

example, Russian Orthodox thinker Sergei Fudel' proposed the concept of the Church's dark twin to describe this phenomenon. In his memoirs written in the Soviet Union in the 1970s, Fudel' tells the story about a priest who announced to his congregation that he was leaving the ministry: "I have deceived you for twenty years, and now I am taking off these vestments". The reaction of the people was "scream[ing], noise, cry[ing]". However, one young man went up to the ambon and said: "After all, it has always been so. Remember that Judas was also sitting at the Last Supper". Then, Fudel' concludes: "These words reminiscent of the existence of a dark twin of the Church in history somehow calmed many or explained something. Moreover, attending the Supper, Judas did not break the sacrament" (Fudel' 2001, pp. 120–23).

We see that the priest's behavior, abnormal and frankly strange from the flock's point of view, led to confusion and anxiety. The main fear was the 'violation of the sacrament', undermining the clerical guarantee to salvation. For the church consciousness colonized by clerical ecclesiologies, the 'integrity' of the clerical structure (the sacrament performed in the assembly of the people) is more important than a person (priest) who dares to act honestly. The restoration of balance occurs through the stigmatization of a strange Other, which "calms many". The priest becomes a Judas who betrayed Christ. Fudel's dark twin is a side of church life that must be fought and condemned. The feeling of a dark twin gives rise to horror. Additionally, the fight against it certainly has clerical features: according to Fudel', a person excommunicates him/herself from the church through involvement in the dark twin, and s/he is reunited with it through confession after the cleric reads the prayer of permission.

Finally, Fudel' concludes that everything distorted, unclean, and wrong that we see in the within the church walls is not a church. This approach does not distinguish between the ethically unacceptable and the simply strange. Both belong to the dark twin. There is also no distinction between sinner and sin, which makes this concept ethically doubtful. Nothing is surprising in Fudel's approach. The concept of the 'dark twin of the church' can be included in the group of ecclesiologies that arose under the influence of the ideas of romanticism. This group includes almost all the modern Orthodox ecclesiological thought from Alexei Khomyakov to Metr. John Zizioulas. For such ecclesiologies, it is typical to regard the church as an integral whole where everything 'dark' (in the above negative sense) either dries up, dies and falls away, or is defined as a disease that must be fought and ultimately destroyed.

Fudel's dark twin is not an approach to describing the abnormal and the strange but a way of excluding it from the church. Additionally, it is a way of demonization of the dark (and automatically of dark actors), which Keller pointed out. Fudel's example shows that there is no language of description for the strange and abnormal in the church, except the negative. Speaking in the metaphor of light described above, normative Orthodox ecclesiologies give too strong a light making it impossible to see the weak luminescent light coming from dark actors. They reduce ecclesiological space to their normative descriptions, in which clerical structures play a predominant role. This in turn leads to the situation of Morton's ouroboros.

## 6. Problem of Light-Striking

Indeed, when trying to talk about the dark, we are constantly faced with the problem of light penetrating—how to talk about the dark while leaving it dark? The light of reason does violence to the dark; it modifies, transforms, deprives it of details, flattens it. Russian philosopher Polina Khanova parallels Jacques Derrida's reflections on how it is possible to speak on behalf of madness (which has no voice) without turning it into another variation of reason (Khanova 2019). In his reflections, Derrida refers to the image of "black light", which he calls the vigil of the 'forces of irrationality' around Cogito. In a sense, this vigil of the forces of irrationality coincides with Morton's mysterious hermeneutic clouds of ignorance. We might say that these 'mysterious clouds' emit 'black light'. Gregory of Nyssa called the divine darkness in which Moses entered luminous darkness.

American cultural theorist Svetlana Boym offers another approach to the problem of light penetrating. She suggests talking about the dark as a game with shadows. Boym makes a subtle distinction between enlightenment and luminosity. She draws on German philosopher Hannah Arendt's reasoning that in dark times, in extreme circumstances, "the illuminations do not come from philosophical concepts but from the 'uncertain, flickering and often weak light' that men and women kindle and shed over the lifespan given to them". The luminous space that emerges is the space of humaneness and friendship, which is "not about having everything illuminated or obscured, but about conspiring and playing with shadows", since friends are always conspirators. The purpose of this light is "not enlightenment but luminosity, not a quest for the blinding truth but only for occasional lucidity and honesty" (Boym 2009–2010).

The light of enlightenment as the light of modern reason always comes from outside, it shines from somewhere *sub specie aeternitatis*. It dominates, objectivates, and compels. The light of which Arendt and Boym write comes from within the human being; it could be compared to the luminescent light that some living organisms emit. It is a gentle light that envelops rather than snatches. Additionally, as the German sociologist of emotions Polina Aronson rightly points out, the opposite of violence is not the absence of violence but tenderness. The Ukrainian theologian Alexander Filonenko argues that "tenderness is attentiveness to the mystery of the Other" (Filonenko 2019). The mystery is impossible without darkness, otherwise, it ceases to be a mystery. John Panteleimon Manoussakis also contrasts violence with caress. His approach allows us to appeal not only to the metaphor of the visual, but also to hearing and touching (Manoussakis 2007). Indeed, luminescent light is not the only thing the dark consciousness perceives; hearing and touching also become ways of knowing in the dark.

## 7. Infra-Language

For better knowledge, we should not rely only on our imperfect abilities to see something in the darkness but also supplement it with the voice of those we try to make out. Thus, by combining two frames of reference (our own and the other's), or rather by constantly moving from one to the other, we can form an irreductionist picture. Bruno Latour calls this approach *infra-language* (Latour 2005). It is a method of studying actors to avoid the risk of rendering all of them mute. To use it, we should constantly move from one system of coordinates to another—from the system of the scholar to the one related with the object. Infra-language is a narrative that requires no privilege for itself, where both the voice of the narrator and the voice of the one being narrated are equally important. According to Latour, this language is fundamentally poor, limited, brief, and simple. It resembles a map rather than a rich landscape. Irreductionism is that we refuse to see the relations between actors as pre-determined. Turning to the metaphor of a map, we can say that we are only given certain points on it, and can then connect them in various ways, not relying on a single normative vision.

In normative ecclesiologies, there is only one system of coordinates set by clerical structures and actors; they are the source of power discourse. At the same time, the rest of the church members have no voice. Sometimes they could be even depersonalized, as exemplified in Miroslav Volf's ironic remark about the ecclesiology of Metropolitan John Zizioulas: his church members are clones of Christ (Volf 1998). Dark ecclesiology gives a voice to the dark actors, intentionally blurring the normativity set by clerical institutions. People define their church membership (in the metaphysical or jurisdictional sense) rather than having someone do it for them. Moreover, church membership can be lived out in various ways, from agnostic doubt to absolute certainty. The normative procedures established by the church hierarchy, such as baptism, etc., are secondary to the witness of the people themselves (unbaptized saints confirm this thesis).

For example, Orthodox LGBTQ activist Misha Cherniak states in the preamble to his article in a collection on Orthodox inclusion: "I am writing this paper not as an academic theologian, but as an Orthodox believer conversant in Orthodox theology and as an activist

who believes that there is always a creative solution and a way out of deadlocks" (Cherniak 2017, p. 141). His article contains not only analytical tools and procedures for making certain inferences but also his personal experience of being a homosexual in the Orthodox Church. It is noteworthy that Cherniak describes his place as a representative of the LGBTQ community in the church, referring to the concepts of apophaticism, oikonomia, and the mystery of divine providence (concepts that traditionally blur the 'light' of theological reason). At the same time, he does not call for the inclusion of LGBTQ in the normative description of the Church. Cherniak's narrative can be interpreted as simply "I am here—inside the Church". This narrative describes his self-definition. In the logic of dark ecclesiology, the theologian cannot ignore this self-determination, even if church authority excludes this actor from the Church. However, it is essential to realize that infra-language involves the description of a particular actor, not the construction of a generalized figure. A homosexual in the Church may be an oppressed actor or an oppressor.

Returning to the metaphor of darkness and light, we can say that Cherniak, as a dark actor, luminesces (as Boym would say), bearing witness to himself. It is noteworthy that in the same collection of articles, there is an article of an anonymous Orthodox bishop who deliberately hides in the shadows (Anonymous 2017; in the Russian edition he is called Anonymous Orthodox bishop, while he is only "Anonymous" in the English version). An irreductionist approach to actors and things assumes that they are all equally important in their existence. In other words, they have an inalienable right to exist.

## 8. Flat Ontology

Since all objects are surrounded by hermeneutic clouds of ignorance, all we can claim about their ontological status is that they equally exist. This principle is called 'flat ontology'. The American OOO-philosopher Ian Bogost describes it as follows: "All things equally exist, yet they do not exist equally" (Bogost 2012, p. 11). Flat ontology implies that being is not hierarchical and no one single object can receive ontological priority. Graham Harman understands it as ontology that "initially treats all objects in the same way, rather than assuming in advance that different types of objects require completely different ontologies" (Harman 2018a, p. 54). He believes that flat ontology "is a useful way of ensuring that we do not cave in to our personal prejudices about what is or is not real" (Ibid., p. 55). For him, flat ontology "is the idea that philosophy must begin by casting the widest possible net in aspiring to talk about everything" (Ibid., p. 256). Another OOO-philosopher, Levi Bryant, argues, that "flat ontology rejects any ontology of transcendence or presence that privileges one sort of entity as the origin of all others and as fully present to itself" (Bryant 2011, p. 245).

Flat ontology challenges the traditional theistic view of God as an initial cause of all things. According to Bryant, "if a God exists, he is not a sovereign like Leibniz's grand architect that designed and produced this world as the best of all possible worlds, but rather is a tinkerer like the rest of us that must contend with the exigencies of other machines* [*Deleuzian-inspired concept of objects]" (Bryant 2014, pp. 115–16). Bruno Latour also points out, that "religion is not about transcendence, a Spirit from above, but all about immanence to which is added the renewal, the rendering present again of this immanence" (Latour 2001, p. 219). He finds a basis for this claim in the Christian dogma of Incarnation of God.

For Harman, Bryant and Latour, an important reference is the thought of Alfred North Whitehead, who, long before the notion of flat ontology, asserted the ontological equality of humans and non-humans. Whitehead's process-theism and the process-theology that emerged from it allow us to apply the idea of flat ontology to God, who is not a divine sovereign but co-creator with the world (Whitehead 1978). Additionally, the idea of the ontological equality of God and other entities can be drawn from the theology of Jürgen Moltmann, who argues that the *kenosis* of God in Christ has lost the ontological nothingness of creation (Moltmann 1993, p. 218).

## 9. The Notion of 'Collective'

The discovery of dark actors raises the question of their inclusion, rights, and representation. Additionally, here, it is necessary to turn to the issue of the common space where all they exist and act. Problematizing the notion of 'society', Bruno Latour suggests replacing it with 'collective' (French–*collectif*) (Latour 2004). A collective is a 'society' that includes non-human living beings and things in addition to human beings. He proposes this notion by rethinking political ecology. It is no coincidence that both Morton and Latour (as well as Donna Haraway's recent works) turn to ecology. Social theorists and philosophers are increasingly asking themselves how to develop an irreductionist theory of society that would include those entities and things that are typically not represented but influence the lives and activities of people, enter into various symbiotic relationships with them, etc. Dark to social theorists, these actors have great 'gravity'. Human society expands to the complex planetary ecosystem. A similar task faces theologians: to describe the Church as a complex ecosystem.

The concept of the collective is designed to overcome the nature/society dualism and to include "natural" objects (non-humans) in politics. From Latour's perspective, the very notion of nature becomes the "speed bump" that political ecology or environmental politics stumbles upon (Latour 2004, p. 4). The fact is that "political ecology has nothing to do with nature" (Ibid., p. 5; for a similar critique of the dualism of nature and society, see Morton 2007). "Nature" is a constructed entity on behalf of which "Science" speaks in the political space. Latour distinguish sciences and Science and defines the latter as "the politicization of the sciences through epistemology in order to render ordinary political life impotent through the threat of an incontestable nature" (Latour 2004, p. 10). Latour's task is to find a fairer redistribution of power in the political space of the *common world*, to which he applies the term *res publica*, endowing it with a broader metaphysical content. To describe the political dimension of the collective—the space of decision-making—Latour proposes the metaphor of a 'parliament' in which an *assembly* of human and non-human beings act. Ontological equality of these entities gives them equal rights to be presented in the collective.

## 10. Reassembling the Church

The term 'collective' can be applied to the Church in two ways. On the one hand, it is possible to speak of an ecclesial collective. On the other hand, the Church could be included as an embedded set in the world collective.

In addition to the objects and actors of normative ecclesiologies, the ecclesial collective includes human and non-human beings and entities hidden in the church shadows (God is among them). Inclusion in the collective presupposes the ontological equality of its members. Of course, the inclusion of animals and other non-human living beings in the church remains controversial, but being in the liminal space, they somehow influence church life by their existence. An attempt at such inclusion is Lynn Townsend White, Jr.'s concept of the democracy of all God's creatures (White 1967).

In dark ecclesiology, dark actors directly or indirectly find their voice, either independently or through their representatives (theologians). This gaining of voice occurs through contextual theologies that introduce discourses reflecting the views and concerns of the shadow groups into the common church space. Direct and independent participation takes place when the voice is directly owned by a representative of the group (women, LGBTQ people, etc.). Indirect participation occurs when the group is represented by those who do not directly belong to it but speak on its behalf (as a rule, we are talking about the representation of non-humans). Eco-theology is an example of a discourse representing those with no independent voice. God also has a voice in the Church, but we have no direct access to him. That is why God's voice sounds through representatives and intermediaries interpreting his will. God's word is recorded mainly in Scripture, which is also recorded through intermediaries and, therefore, cannot be considered a straightforward expression of his will.

The communication space for decision-making in the church collective can be described through the metaphor of a *sobor*. I intentionally use a term unaccustomed to the anglophone space in order to avoid unnecessary associations. Just as sobornost' transcends conciliarity or synodality (Valliere 2012, p. 11), sobor is not simply a council or a synod, consisting mainly of those who are "in the light". It is closer to an assembly in Latour's sense.

The sobor includes different factions, representing the interests of various groups and individual beings. It is not an institution but a dynamic process of relating different perspectives in the space of the Church, which ultimately changes its view of itself. For example, in one historical period, a phenomenon may qualify as a sin that prevents salvation in eternity, while it can become a norm in another one. For example, some Christian communities no longer consider homosexuality a sin and accept this format of relationships as a church norm, sanctifying same-sex marriages in the sacrament of marriage. Likewise, the attitude toward non-human beings is changing. In the encyclical *Laudato Si*, Pope Francis reconsiders the traditional Catholic instrumental attitude toward animals, for example, recognizing their intrinsic value for God.

Just as in *res publica*, the unifying principle of the collective is the covenant of rules for living together, accepted by all participants in a common political space. In the Church, the unification of the ecclesial collective takes place through the covenant of God and church members. The covenant presupposes the equality of its participants and the free acceptance of the terms of the covenant. This covenant does not imply static norms, given once and for all, but a dynamic process of the sobor of the church collective constantly redefining norms.

One of the first examples of the sobor's work in Christian Church history was the inclusion in the Church of former Gentiles who had not undergone circumcision. The discussion was about the relationship of circumcision to salvation, with part of the early Christian community of Christ's disciples believing that salvation was impossible without it. For them, non-Jewish (Gentile) converts who had not accepted circumcision were initially in the Church's shadow but found their voice through the representation of the apostles Paul, Peter, etc. (Acts 10–11, 15). The Book of Acts also describes God's participation in this process (Acts 10:3–20), but his voice was not that of a church sovereign. As a result, the refusal to require circumcision became part of the New Testament, but after Christ's ascension. The new norm developed with the voice of God in mind, but it required discussion among the apostles and was ultimately adopted by them. The account of the acceptance of former Gentiles into the Church later became paradigmatic for other groups, such as homosexuals (Perry 2010).

In the history of the Church, God does not act as a sovereign. His voice can be heard through his representatives but is often lost among other voices and sometimes even distorted by intermediaries to strengthen their own power. Theologically, this "weakness of God" has to do with providing freedom to human beings, a process in which God does not want to interfere by force because of His love for humanity (see more in Caputo 2006). However, God does not abandon humanity and His Church, and so participates in the work of the church collective together with others in equality. God's kenotic love can be explained as the possibility of His self-denial before the creativity of the creatures He has created, who are deciding their own destiny (that is why He looks like a tinkerer).

In the space of the common world, God and other spiritual beings play no less a role than non-human living beings belonging to the "natural world" since politics is done on their behalf. Additionally, while Latour incorporates the latter into the world collective through political ecology, which overcomes the dualism of society and nature, the former must be incorporated through political theology.

The task of dark theology here is to identify those who claim to speak for God in the public space, analyze and discern the ways and formats of this representation, and identify and critique strategies for appropriating power in God's name.

**Funding:** This research was funded by the Estonian Research Council grant number PUT 1599. And The APC was funded by Henry Luce/Leadership 100: Orthodoxy and Human Rights.

**Conflicts of Interest:** The author declares no conflict of interest.

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
