# Peer review of "Dark Theology as an Approach to Reassembling the Church"

_religions, doi:10.3390/rel13040324_

Round 1

Reviewer 1 Report

This is an excellent article.  Creative, constructive, and original. 

 I've been doing this for a very long time, and I know good Orthodox theology when I see it.  

As I said, this is "original, creative, and constructive."  It advances a thesis that is arguable and yet to be seen.  It speaks from a particular perspective.  It's an article that should be published and read as it advances the conversation. Especially given the situation in Ukraine, it is extremely timely and relevant. 

Author Response

Thank you so much for the review!

Reviewer 2 Report

Great paper!

Author Response

Thank you so much for the review!

Reviewer 3 Report

Minor issue: Line 28 should be (et al) and not (etc)

Overall, I found the second half of this paper to be very strong. The main area that needs attention is the first half of the paper which presents and situates "dark theology" in a way that needs revision and expansion. 

In the early paragraphs discussing the two extremes dark theology is trying to avoid, it might be worthwhile to frame this as Marxian social reductionism on the one hand, and Platonic idealist metaphysics on the other. This is helpful in identifying the roots of the trajectories of these two extremes the author is trying to avoid.

It is a bit unclear as to whether the author originated "dark theology" or whether it comes from elsewhere. It is not a widely known term, and the author needs to indicate where it emerged from rather than simply stating it emerged from debates on Orthodoxy and human rights. Where did these debates occur? Is this in published articles? Conferences? Councils? Online forums? While this is touched on in the middle of the article, it would be helpful to have a summary paragraph on the topic early in the article. 

MOST SIGNIFICANT AND NECESSARY REVISION: Lines 185-204 are problematic reading of Dionysius and mystical theology. Dionysius and Gregory of Nyssa are adamant that the encounter with God by Moses occurs in the divine darkness and that this is where we too encounter God.  Apophatic theology as such is merely a stage in the flow of mystical theology, which itself sounds almost identical to “dark theology”. I.e. the tripartite flow of mystical theology is almost identical to what the author calls "dark theology." Apophasis is one of the theological stages of the mystical move.

The author may want to address this by framing things in two ways: (1) identify that apophatic theology divorced from the full flow of the mystical theology of Dionysius is problematic, but that the mystical theology of St. Gregory and St. Dionysius is quite close to "dark theology"; and (2) should perhaps consider defining “dark theology” more simply as mystical theology applied to social issues and embodiment.

The author (and paper) would also benefit from taking a look at Simon Critchley’s “Mystical Anarchism,” Critical Horizons: A Journal of Philosophy and Social Theory 10(2), August 2009, 272–306, and incorporating it into the argument. This article engages Carl Schmitt, political theology, and the liberatory embodiment of medieval mystical "anarchists" such as Marguerite Porete and her feminist embodiment of mystical theology.

Excellent critique of clerico-centrism. This section could be expanded and backed by further examples, although this is not necessary.

Lines 515-516 invoke John D. Caputo but do not cite him. This should be rectified by acknowledging Caputo's emphasis on the weakness of God and pioneering "weak theology." 
